# Self-Adaptive Fine-grained Multi-modal Data Augmentation for Semi-supervised Muti-modal Coreference Resolution

Li Zheng*
zhengli@whu.edu.cn
School of Cyber Science and
Engineering, Wuhan University

Boyu Chen*
chenboyu1225@whu.edu.cn
School of Cyber Science and
Engineering, Wuhan University

Hao Fei
haofei37@nus.edu.sg
School of Computing, National
University of Singapore

Fei Li*†
foxlf823@gmail.com
School of Cyber Science and
Engineering, Wuhan University

Shengqiong Wu
swu@u.nus.edu
School of Computing, National
University of Singapore

Lizi Liao
lzliao@smu.edu.sg
SCIS, Singapore Management
University

Donghong Ji*
dhji@whu.edu.cn
School of Cyber Science and
Engineering, Wuhan University

Chong Teng*
tengchong@whu.edu.cn
School of Cyber Science and
Engineering, Wuhan University

## ABSTRACT

Coreference resolution, an essential task in natural language processing, is particularly challenging in multi-modal scenarios where data comes in various forms and modalities. Despite advancements, limitations due to scarce labeled data and underleveraged unlabeled data persist. We address these issues with a self-adaptive fine-grained multi-modal data augmentation framework for semi-supervised MCR, focusing on enriching training data from labeled datasets and tapping into the untapped potential of unlabeled data. Regarding the former issue, we first leverage text coreference resolution datasets and diffusion models, to perform fine-grained text-to-image generation with aligned text entities and image bounding boxes. We then introduce a self-adaptive selection strategy, meticulously curating the augmented data to enhance the diversity and volume of the training set without compromising its quality. For the latter issue, we design a self-adaptive threshold strategy that dynamically adjusts the confidence threshold based on the model's learning status and performance, enabling effective utilization of valuable information from unlabeled data. Additionally, we incorporate a distance smoothing term, which smooths distances between positive and negative samples, enhancing discriminative power of the model's feature representations and addressing noise and uncertainty in the unlabeled data. Our experiments on the widely-used CIN dataset show that our framework significantly outperforms state-of-the-art baselines by at least 9.57% on MUC F1 score and 4.92% on CoNLL F1 score. Remarkably, against weakly-supervised baselines, our framework achieves a staggering 22.24% enhancement in MUC F1 score. These results, underpinned by in-depth analyses, underscore the effectiveness and potential of our approach for advancing MCR tasks.

## CCS CONCEPTS

• **Computing methodologies → Artificial intelligence**.

## KEYWORDS

Coreference Resolution, Multi-modal, Semi-supervised Learning

## 1 INTRODUCTION

Coreference resolution (CR), which identifies all mentions referring to the same entity, plays a crucial role in many downstream applications, such as relation extraction [11, 49], question answering [5, 46, 55], and sentiment analysis [15, 24, 53, 56, 57]. However, in real-world scenarios, particularly in social media, data often comes in various forms and modalities [16, 28, 29, 43, 44], including text, images, rather than pure text. Therefore, some recent advancements have transferred to multimodal coreference resolution [17, 18], where coreference occurs not only between the entities in text but also the objects in image, as shown in Figure 1 (a). Compared to the text-based coreference resolution, Multimodal Coreference Resolution (MCR) presents even greater challenges due to the substantial semantic gap between different modalities and the scarcity of annotated data.

Existing works have made commendable efforts in MCR. Goel et al. [18] proposed a multimodal pipeline and utilized weak supervision to identify coreference chains. In contrast, another study by Goel et al. [17] argued that weak supervision alone is unable to resolve the ambiguity of multiple instances of the same object class. To address this issue, they introduced semi-supervised learning and employed cross-modal attention to integrate image region features with textual features. Despite these advances, current methods

*Key Laboratory of Aerospace Information Security and Trusted Computing, Ministry of Education
†Fei Li is the corresponding author.

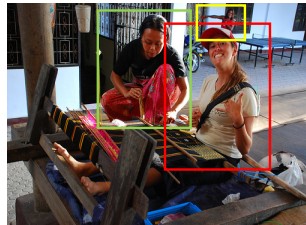

in front of the image there is a person wearing a cap and she is sitting on the blanket. there is a cloth on the wooden object. there is another person holding a wooden stick in her hand. there are a few objects in a cover. there are some objects. there are doors. there are windows on the wall. in the background of the image there is a person holding some object in her hand. there is a net on the tables. there is a helmet on the bike.

(a) an example from the CIN dataset

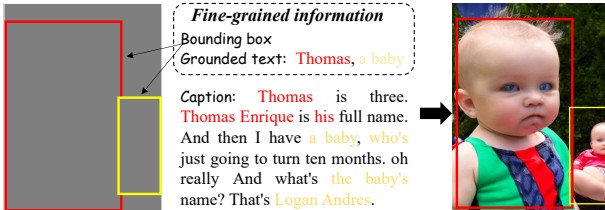

(b) an example of fine-grained multimodal data generation

**Figure 1: An example from the CIN dataset (a) [18] and the process of fine-grained multimodal data generation (b).**

in MCR still tied to model optimization and extracting information features. Data, as a key dimension that can significantly affect model performance, has been far less explored. Therefore, we summarize the challenges facing current research from a data-centric perspective in two aspects:

• **Scarcity of labeled data.** The manual annotation of large-scale fine-grained multimodal datasets is costly, resulting in a scarcity of extensively labeled data required for research on multimodal coreference resolution. This scarcity poses significant challenges in training MCR models. Currently, automatic data expansion [8, 23] is widely used as a practical technique to alleviate the data scarcity problem. Xiao et al. [47] applied the stable diffusion [35] to generate high-quality images with a given text input to solve the image captioning task, but the method is limited to coarse-grained expansion of text-image pairs, ignoring fine-grained objects, entities, etc. For the coreference resolution task, existing methods [10] mainly focus on textual coreference resolution data augmentation, while the MCR data augmentation remains largely unexplored. Therefore, how to adaptively expand high-quality fine-grained multimodal labeled data is the essence of successful MCR.

• **Under-exploitation of unlabeled data.** On the other hand, despite addressing the fundamental issue of scarcity of labeled data through adaptive data augmentation, the untapped potential of unlabeled data remains underexplored. In semi-supervised learning, existing methods [17] set a fixed threshold to ensure the quality of pseudo-labels for unlabeled data. Unfortunately, a fixed high threshold (e.g., 0.9) poses several limitations: (1) Loss of useful information: By using a fixed high threshold, many unlabeled data that provide valuable insights are filtered out, potentially discarding useful samples. (2) Limitation on the quantity of training data: With a fixed high threshold, only a limited number of unlabeled data are selected, restricting the scale of training data. (3) Ignoring the model's learning status: A fixed threshold ignore the dynamic nature of training process as it cannot dynamically adjust the threshold based on the model's learning status. Therefore, it is crucial for the MCR task to

effectively harness valuable information from unlabeled data based on the learning status of the model in order to train more robust and accurate MCR models.

Taking into consideration the two aforementioned aspects, we propose a novel framework: *Self-adaptive Labeled and Unlabeled multimodal Data Augmentation (SLUDA)*, to improve semi-super vised MCR. **Firstly**, as depicted in Figure 1 (b), we utilize a text CR dataset [32] and a diffusion model [27], as well as randomly generated bounding boxes, to generate text-image pairs and object-entity pairs for fine-grained data expansion. Subsequently, we design a self-adaptive selection strategy to filter and select the generated multimodal data and combine it with existing labeled data for model training. This enables us to increase the quantity and diversity of data while ensuring quality, achieving the first goal of data augmentation for labeled data. **Next**, we adopt a self-adaptive threshold strategy to fulfill the second goal of fully tapping into the unlabeled data. By dynamically adjusting the confidence threshold based on the model's learning status and performance, we estimate the threshold based on the exponential moving average (EMA) of confidence scores from unlabeled data. This allows for the utilization of a lower threshold during the early stage of training, and helps the model to accelerate convergence. As training progresses, the threshold gradually increases to filter out unreliable pseudo-labels and enhance the accuracy of the model. **Furthermore**, considering that pseudo-labeled data are unavoidably noisy than manually-labeled data, we introduce a distance smoothing term. It smooths the distances between positive and negative samples, aiding the model in learning more discriminative feature representations and improving the utilization of unlabeled data.

To verify the effectiveness of our model, we conduct experiments on the benchmark CIN dataset [18]. The results demonstrate that our framework significantly helps the state-of-the-art model to further improve its performance by at least 9.57% on MUC F1 score and 4.92% on CoNLL F1 score. Most strikingly, compared to the weakly-supervised baselines, ours outperforms their MUC F1 score by 22.24%. Further ablation experiments demonstrate that each component of our framework is essential. Additionally, we conduct extensive experiments to demonstrate the effectiveness of adaptive selection and utilization of labeled and unlabeled data in our framework. Moreover, we focus on smoothing distance differences between positive and negative samples, promoting more accurate and reliable feature representation learning. Our main contributions are summarized as follows:

- We propose self-adaptive expansion and selection techniques to enrich the training data with generated high-quality MCR data, effectively addressing the issue of labeled data scarcity.

- We devise a self-adaptive threshold strategy to leverage unlabeled data, achieving a balance between quality and quantity. Moreover, we introduce a distance smoothing policy to enhance the discriminative ability of feature representations.

- Our extensive experimental results on the CIN dataset demonstrate that our SLUDA framework achieves the state-of-the-art performance and outperforms the best baseline with large margins.

## 2 RELATED WORKS

### 2.1 Multimodal Coreference Resolution

Within the multimodal learning community [12–14, 26, 42, 45, 54], there has been growing interest in multimodal coreference resolution. Several studies [17–19] have been conducted on this topic. Goel et al. [18] introduced the CIN dataset, which addresses the problem of coreference resolution in long narratives within visual scenes. They proposed a multimodal pipeline and utilized weak supervision to learn to identify coreference chains. Goel et al. [17] proposed a semi-supervised training approach based on a fixed threshold to effectively learn from unlabeled sets. However, despite achieving partial success, these methods suffered the issue of scarce labeled data and failed to fully leverage the potential of unlabeled data.

### 2.2 Semi-supervised Learning

Semi-supervised learning is an effective paradigm that utilizes a large amount of unlabeled data along with limited labeled data. Various methods [9] have been proposed in the field of semi-supervised learning, such as pseudo-labeling and consistency regularization. In pseudo-labeling, the models [34, 48] use unlabeled samples with high confidence as training targets, thereby reducing the density of data points at the decision boundary. In consistency regularization, the models [2, 4] are self-supervised on unlabeled data, providing additional training signals. To mitigate the confirmation bias in pseudo-labeling, Goel et al. [17] proposed a threshold-based technique to ensure the quality of pseudo-labels, where only unlabeled data with confidence above the threshold are retained. While promising results have been achieved, a fixed high threshold can lead to the disregard of a large number of ambiguously predicted unlabeled examples, especially in the early and mid-training stages.

### 2.3 Multimodal Data Augmentation

Multimodal data augmentation is the process of creating additional training data with high quality and diversity. It has been widely applied in various machine learning tasks [1, 20, 50, 58]. Recently, Wang et al. [40] proposed a framework for paired cross-modal data augmentation, which generates an infinite amount of paired data to train cross-modal retrieval models. Xiao et al. [47] utilized stable diffusion models for text-to-image generation, expanding the training set with high-quality image-caption pairs. In this paper, we employ a diffusion model [27] to augment fine-grained labeled data, addressing the issue of scarce labeled data, enhancing the diversity and richness of the MCR dataset, and improving the generalization and performance of MCR models.

## 3 METHODOLOGY

In this paper, we propose a Self-adaptive Labeled and Unlabeled multimodal Data Augumentation (SLUDA) framework that fully utilizes data from two perspectives to address the MCR task. The architecture is shown in Figure 2. First, we generate fine-grained image-caption pairs and adaptively selects high-quality generated MCR data for labeled data augmentation. Then, we design a self-adaptive threshold strategy, considering factors such as data quality and model confidence, and introduce a distance smoothing term

to enhance the utilization of unlabeled data. Finally, with preprocessed multimodal labeled and unlabeled data, we employ two modality-specific Variational Autoencoders (VAEs) to learn latent representations for different modalities and perform the MCR task.

### 3.1 Task Definition

Let $(I, C)$ represent an image-caption pair, where $C$ describes the image $I$ as shown in Figure 1 (a). We denote the set of $p$ mentions as $M = \{m_1, m_2, ..., m_p\}$, and the image $I$ contains $q$ regions denoted as $I = \{r_1, r_2, ..., r_q\}$. The objective of multimodal coreference resolution is to identify text mentions that refer to the same entity and associate each mention with a specific region in the image.

### 3.2 Labeled Data Augmentation

In the task of multimodal coreference resolution, a large number of fine-grained annotated image-caption pairs are typically required. Existing MCR datasets, such as CIN [18], demand human annotators to label entity mentions referring to the regions in the referenced image, construct coreference entity chains, and draw bounding boxes in the image. This annotation process is not only labor-intensive but also time-consuming. Furthermore, the collected images and annotated mentions may suffer from incompleteness and lack diversity, which limits the generalization capabilities of models trained on such datasets. To address the challenge of scarce labeled data, we propose a novel approach that leverages existing text-based coreference resolution datasets in conjunction with a diffusion model to generate images, and then adaptively select high-quality multimodal data. By doing so, we mitigate the limitations posed by data scarcity in the MCR task.

**Labeled Data Generation.** The diffusion models [36, 47, 51] have been proposed and applied for data augmentation, which is a generative model that generates diverse samples by iteratively diffusing noise signals. For the task of grounded text-to-image generation, Li et al. [27] introduced GLIGEN, which builds upon existing pre-trained text-to-image diffusion models and extends their capabilities to incorporate grounding inputs as conditions. We employ GLIGEN and leverage the widely-used text coreference resolution dataset English OntoNotes 5.0 [32] as the foundational dataset for data augmentation. Given the text as the caption $C = \{w_1, w_2, \ldots, w_n\}$, a set of mentions $M = \{m_1, m_2, \ldots, m_p\}$, and their randomly generated corresponding bounding boxes $B = \{b_1, b_2, \ldots, b_q\}$ as input, we aim to generate the image $I$:

$$I = GLIGEN(C, M, B),\qquad(1)$$

where $I = \{r_1, r_2, ..., r_q\}$ represents the image $I$ with $q$ regions. We discard images that contain NSFW (Not Safe for Work) content. Then, we pair the generated images with the corresponding caption to form an augmented labeled dataset.

**Self-adaptive Selection.** Due to the uncertain quality of the generated images, we employ a self-adaptive selection strategy to filter the data based on the quality assessment of the image-caption pairs, ensuring both the quality and quantity of the training set. Specifically, since CLIPScore [21] is used to assess whether the generated image can match and correlate with the given textual description, we evaluate the generated image-caption pairs using

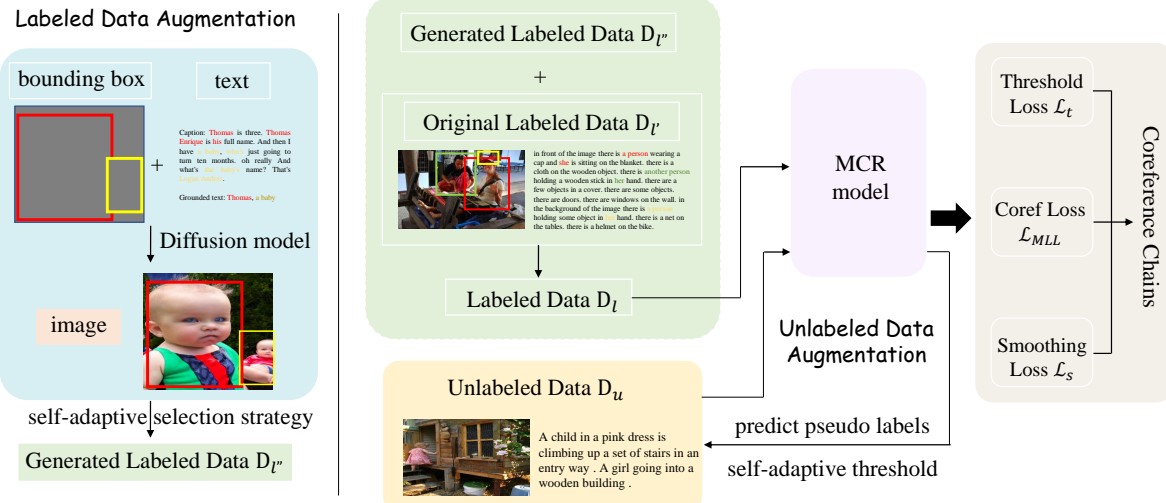

**Figure 2: The overall architecture of our model. Initially, we employ a diffusion model to expand the fine-grained MCR labeled data and design a self-adaptive selection strategy to filter and obtain the generated labeled data $D_{l''}$. Subsequently, $D_{l''}$ is merged with the original labeled data $D_{l'}$, and they are combined with the unlabeled data $D_u$ as input to the MCR model. Then, a self-adaptive threshold strategy is adopted to fully tap into the valuable information of unlabeled data.**

CLIPScore and sort them according to their evaluation scores. During the initial stages of training, we incorporate all the generated data as labeled data into the training set to ensure sample diversity. As training progresses, we filter out a portion of low-scoring generated data based on the model's learning status to maintain the quality of the labeled data. The learning status of the model is estimated using the predicted scores, which are then smoothed using Exponential Moving Average (EMA) [31]. EMA calculates the current score as a weighted average of the current and previous EMA values, providing a more stable reflection of the model's learning status. The filtering ratio $\tau_{r_t}$ is defined as follows:

$$EMA = \alpha_1 * o_{r_t} + (1 - \alpha_1) * \tau_{r_{t-1}}, \tau_{r_t} = exp(-\beta * EMA), \quad (2)$$

where $\alpha_1$ is the smoothing factor ($0 \sim 1$), $o_{r_t}$ represents the prediction accuracy at time $t$, $\beta$ serves as a control parameter for the filtering intensity. When the EMA value is higher, indicating better overall predicted scores and superior performance of the model in handling the task, we filter out a larger proportion of generated data to ensure the quality of labeled data. Conversely, when the model's learning status is poorer, we filter out a smaller amount of generated data, allowing for the retention of a larger number of samples for training. This approach enables a balance between the quality and quantity of the training set throughout the training process, effectively leveraging the labeled data to enhance the model's performance. This generated and adaptively filtered labeled dataset is denoted as $D_{l''}$, which is merged with the labeled dataset $D_{l'}$ from the multimodal dataset CIN. Therefore, the final labeled dataset is $D_l = D_{l''} + D_{l'}$.

### 3.3 Unlabeled Data Augmentation

From the perspective of unlabeled data, existing method [17] fails to fully utilize the potential of unlabeled data, resulting in the waste of unlabeled data and a decrease in the robustness of MCR task. Moreover, unlabeled data often contains noise, and reducing the

impact of noise on the model while promoting more accurate and reliable feature representation learning is crucial for MCR. Thus, we propose self-adaptive threshold and distance smoothing techniques to effectively leverage unlabeled data and mitigate noise within it.

**Self-adaptive Threshold.** Considering that the fixed high threshold method results in the wastage of numerous samples, the samples with ambiguous predictions contribute negligibly during the training phase. Hence, to harness unlabeled data effectively, we propose a self-adaptive threshold strategy that reflects the learning status of the model. The strategy estimates the effectiveness of learning based on the model's prediction confidence and subsequently calculates the threshold $\tau_t$ using the exponential moving average of the confidence. The threshold $\tau_t$ is defined and adjusted as follows:

$$\tau_t = \alpha_2 * o_t + (1 - \alpha_2) * \tau_{t-1}, \quad (3)$$

where $t$ represents the time step of training, $o_t$ represents the prediction accuracy at time $t$, and $\alpha_2$ is the smoothing factor ($0 \sim 1$). By employing this strategy, we can use a lower threshold in the early stages to leverage unlabeled data and accelerate the model convergence. As the training progresses, we gradually increase the threshold to filter out unreliable pseudo-labels and improve the model's accuracy. For unlabeled data, we focus on using cross-entropy loss with a confidence threshold for entropy minimization-based pseudo-labeling:

$$\mathcal{L}_t = -\sum_{m \in N} \sum_{r \in I} \mathbb{I}(p(y|(m,r)) > \tau_t) log(p(y|(m,r))), \quad (4)$$

where $p(\cdot)$ is the output probability from the model and $\mathbb{I}$ is the indicator function for confidence-based thresholding with $\tau$ being the threshold.

**Distance Smoothing.** To address the issue of noise in unlabeled data, we introduce a distance smoothing term to smooth the distances between positive and negative samples. This approach aims to facilitate the model's ability to learn more discriminative

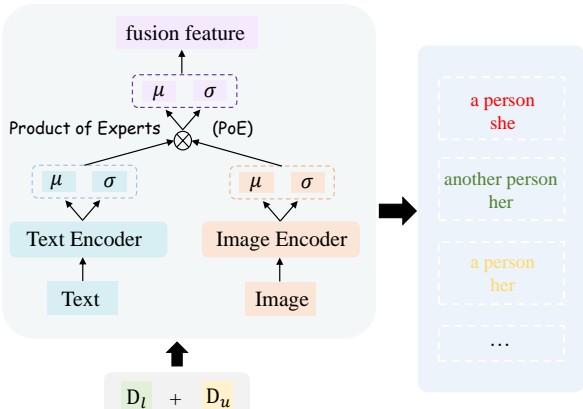

**Figure 3: The details of our MCR model.**

feature representations and maximize the utilization of unlabeled data. First, we compute the cosine similarity scores between each pair of samples to determine whether they are positive or negative samples. For each mention $m$, we compare its similarity score with another mention $m'$ and use a predefined threshold to determine their labels. If the similarity score exceeds the threshold, it is labeled as a positive sample. Otherwise, it is labeled as a negative sample. Next, for each pair of positive samples $(m_i, m_j)$, we calculate their Euclidean distance $d_{ij} = ||m_i - m_j||$. Similarly, for each pair of negative samples $(m_i, m_k)$, we calculate their Euclidean distance $d_{ik} = ||m_i - m_k||$. To smooth the distance differences between positive and negative samples, we introduce a smoothing function $g(d_{ij}, d_{ik})$ defined as:

$$g(d_{ij}, d_{ik}) = 1/(1 + exp(-\gamma * (d_{ij} - d_{ik})))) . \qquad (5)$$

This smoothing function maps the distances between positive and negative samples to a smooth value between 0 and 1, providing a smoothing effect by reducing the distance disparity. Finally, we incorporate the distance smoothing term into the loss function to encourage the model to learn discriminative feature representations and effectively utilize unlabeled data. The formula for the loss function is as follows:

$$\mathcal{L}_s = \sum_{i=0}^{p} \left[ \frac{1}{|\mathcal{P}|} \sum_{j \in \mathcal{P}} d_{ij}^2 + \frac{1}{|\mathcal{N}|} \sum_{k \in \mathcal{N}} max(0, \lambda - d_{ik})^2 \right] * g(d_{ij}, d_{ik}) ,$$
$$(6)$$

where $\mathcal{P}$ is the positive set, $\mathcal{N}$ is the negative set. The parameter $\lambda$ controls the separation between positive and negative samples. We design this loss function to narrow the gap between positive examples and widen the gap between positive and negative examples. The smoothing function $g(d_{ij}, d_{ik})$ has a value close to 1 when the difference between $d_{ij}$ and $d_{ik}$ is large, increasing the difference in distance between positive and negative samples. When the difference between $d_{ij}$ and $d_{ik}$ is small, the value of $g(d_{ij}, d_{ik})$ is close to 0, decreasing the distance difference between positive and negative samples. By employing the distance smoothing method, we are able to smooth the distance disparities between positive and negative samples, reduce the impact of noise on the model, and facilitate more accurate and reliable feature representation learning, thereby enhancing the model's performance.

## 3.4 MCR Model

The details of our model are shown in Figure 3. For fair comparison, following Goel et al. [17], we utilize BERT [7] as the text encoder to obtain contextualized word representations and Faster RCNN [33] as the visual encoder to obtain region representations of images. To effectively integrate textual and visual features, we employ two modality-specific modality-specific Variational Autoencoders (VAEs) to obtain latent representations of the two modalities' features. Then, by applying Product of Experts (PoE) [22] to the latent representations of both modalities, we obtain the multimodal representation.

**Text Feature Extraction.** We employ BERT to encode each word in the narrative caption $C = \{w_1, w_2, \ldots, w_n\}$, as mentioned in Goel et al. [17]. We prepend a [CLS] token and append a [SEP] token to each sentence, resulting in $C_0 = \{[CLS], w_1, w_2, \ldots, w_n, [SEP]\}$. We then concatenate them together as the input to BERT to generate contextualized token representations. The mention embeddings $M = \{m_1, m2, \ldots, m_p\}$ are computed by averaging the corresponding word embeddings.

**Image Feature Extraction.** Following Goel et al. [17], we employ Faster RCNN to extract region representations of the image $I = \{r_1, r_2, \ldots, r_q\}$. Each region $r_i$ is represented by a d-dimensional joint embedding $r_i \in \mathbb{R}^d$, which encompasses its visual, semantic, and spatial features.

**Multimodal Feature Fusion.** Previous research [17] has treated text and image features equally, mapping the concatenated features of both modalities to the same latent representation. However, there exist mismatches between text and image, introducing noise into the models used for prediction. To address this, we utilize modality-specific VAEs to map the features of the two modalities into their respective latent representations with internally correlated distributions. This enables the model to adaptively learn its specific data representation for each modality, better capturing the intrinsic relationships and structures among the features of each modality, helping to address the mismatch between text and image. The encoder of each VAE includes dense layers that map the input features to mean vectors $\mu$ and standard deviation vectors $\sigma$.

For the text modality, the text feature $c$ is inputted into the encoder of the text-VAE to parameterize the mean vector $\mu_c$ and the standard deviation vector $\sigma_c$. We can approximate the true posterior distribution $p(z_c|c)$ using these parameters, where the distribution of $z_c$ can be represented as $z_c \sim q(z_c|c) = N(\mu_c, \sigma_c^2)$. For the visual modality, the image feature $r$ is also inputted into the encoder of the image-VAE to obtain an approximation of the true posterior distribution $p(z_r|r)$, where the distribution of $z_r$ can be represented as $z_r \sim q(z_r|r) = N(\mu_r, \sigma_r^2)$. Based on the assumption of conditional independence of multimodal latent representation, the latent distribution $p(z_m|c, r)$ of the multimodal representation can be simplified as the combination of two separate latent distributions $p(z_m|c)$ and $p(z_m|r)$. Thus, we apply the product-of-experts (PoE) to estimate the multimodal latent distribution:

$$p(z_m|c, r) \propto p(z_m|c)p(z_m|r) = q(z_c|c)q(z_r|r) , \qquad (7)$$

We assume that the latent representations are independent and follow Gaussian distributions parameterized by mean and standard deviation. Therefore, the distribution of $z_m$ can be represented as

**Table 1: Coreference resolution results on the CIN dataset [18] from our proposed method and other state-of-the-art multi-modal baselines. ∗ means zero-shot performance and others are semi-supervised performance. In the brackets are the improvements of our model over the best-performing baseline(s).**

| Method | MUC | | | $B^3$ | | | $CEAF_e$ | | | CoNLL |
|---|---|---|---|---|---|---|---|---|---|---|
| | R | P | F1 | R | P | F1 | R | P | F1 | F1 |
| • *Text-based Methods* | | | | | | | | | | |
| Rule-Based [25]∗ | 5.60 | 10.13 | 6.40 | / | / | / | / | / | / | / |
| Neural Coref [25]∗ | 0.11 | 0.17 | 0.13 | / | / | / | / | / | / | / |
| longdoc [38]∗ | 7.79 | 8.43 | 7.24 | 62.27 | 76.10 | 67.69 | 48.77 | 84.95 | 61.02 | 45.31 |
| • *Multimodal Methods* | | | | | | | | | | |
| VisualBERT [37]∗ | 18.17 | 6.08 | 8.06 | 69.01 | 36.08 | 41.03 | 21.25 | 57.10 | 28.67 | 25.92 |
| UNITER [6]∗ | 16.92 | 7.15 | 8.83 | 68.34 | 44.29 | 50.22 | 28.12 | 72.78 | 38.91 | 32.65 |
| VinVL [52]∗ | 16.76 | 8.60 | 9.75 | 68.49 | 62.32 | 61.30 | 42.88 | 80.81 | 53.69 | 41.58 |
| MAF [41] | 19.07 | 15.62 | 15.65 | / | / | / | / | / | / | / |
| WS-MCR [18] | 24.87 | 18.34 | 19.19 | / | / | / | / | / | / | / |
| Semi-MCR [17] | 31.11 | 35.25 | 31.86 | 70.63 | 87.85 | 78.06 | 63.99 | 93.44 | 75.47 | 61.79 |
| **Ours** | **39.83** | **44.70** | **41.43** | **72.42** | **90.27** | **80.32** | **66.51** | **96.63** | **78.38** | **66.71** |
| | (+8.72%) | (+9.45%) | (+9.57%) | (+1.79%) | (+2.42%) | (+2.26%) | (+2.52%) | (+3.19%) | (+2.91%) | (+4.92%) |

**Table 2: Comparison of narrative grounding performance on the CIN dataset. In the brackets are the improvements of our model over the best-performing baseline(s).**

| Method | Noun Phrases | Pronouns | Overall |
|---|---|---|---|
| MAF [41] | 21.60 | 18.31 | 20.91 |
| WS-MCR [18] | 30.27 | 25.96 | 29.36 |
| Semi-MCR [17] | 32.58 | 28.45 | 31.71 |
| **Ours** | **36.84** | **32.14** | **35.72** |
| | (+4.26%) | (+3.69%) | (+4.01%) |

$z_m \sim N(\mu_m, \sigma_m^2)$, where $\mu_m = \frac{\mu_c \sigma_r^2 + \mu_r \sigma_c^2}{\sigma_c^2 \sigma_r^2}$ and $\sigma_m^2 = (\frac{1}{\sigma_c^2} + \frac{1}{\sigma_r^2})^{-1}$. The latent variable $z_m$ of the multimodal representation can be computed as $z_m = \mu_m + \sigma_m^2 \odot \epsilon$ , where $\epsilon \sim N(0, I)$.

**Training.** After obtaining preprocessed multimodal labeled and unlabeled data, we feed them into our model to learn latent representations of different modalities and perform coreference resolution. We calculate the score for the multimodal mention pair $(z, z')$ as $S_{mcr} = \frac{z \cdot z'}{|z||z'|}$. This score measures the likelihood of coreference between $z$ and $z'$. A higher score indicates a stronger indication of coreference, whereas a lower score suggests non-coreference. We train the coreference scorer by utilizing the negative log marginal likelihood ($\mathcal{L}_{MLL}$). Finally, the total loss function is the summation of all the aforementioned losses:

$$\mathcal{L} = \mathcal{L}_{MLL} + \eta_1 \mathcal{L}_s + \eta_2 \mathcal{L}_t . \tag{8}$$

## 4 EXPERIMENTS

### 4.1 Experimental Setting

**Datasets.** We evaluate our approach on the CIN dataset [18], which consists of 1,000 test and 880 validation image-caption pairs. Following Goel et al. [17], we use the annotations from the validation split of the CIN dataset as a small labeled set during training. To augment our data, we utilize the widely used text coreference resolution dataset English OntoNotes 5.0 [32], which contains 3,493 documents. After carefully filtering the documents, we obtain 3,116 multimodal coreference examples as labeled data.

**Evaluation Metrics.** In terms of evaluation metrics, we align with Goel et al. [17] and use CoNLL F1 for coreference resolution evaluation, which is the average F1 score across three metrics: MUC [39], $B^3$ [3], and $CEAF_e$ [30]. For narrative grounding, we consider a prediction to be correct if the Intersection over Union (IoU) between the predicted bounding box and the ground truth box is greater than 0.5 [18].

### 4.2 Baseline Systems

To validate the effectiveness of our model, we compare it against the following state-of-the-art baselines:

- **VL-BERT:** Su et al. [37] developed VL-BERT with rich aggregation and alignment functions.

- **UNITER:** Chen et al. [6] introduced UNITER, which uses Optimal Transport to align words to image regions.

- **VinVL:** Zhang et al. [52] developed an object detection model to provide object-centric representations of images.

- **MAF:** Wang et al. [41] leveraged fine-grained visual representations to model phrase-object correlation.

- **WS-MCR:** Goel et al. [18] employed weak supervision to learn to identify coreference chains.

- **Semi-MCR:** Goel et al. [17] proposed a semi-supervised approach and designed a multimodal fusion model for MCR.

### 4.3 Coreference Resolution Results

We comprehensively compare our ALUDA against text-based coreference and multimodal coreference methods on the CIN dataset. The results, presented in Table 1, highlight the superiority of our method over the state-of-the-art (SoTA) baselines and reveals several key findings. Firstly, compared to traditional text-based coreference methods, multimodal methods consistently demonstrate superior performance by leveraging additional visual features. But without carefully incorporating visual information into the task, such as VisualBERT, UNITER and VinVL models, they only achieve marginal

**Table 3: Ablation results. The numbers in the brackets are the decreased values compared with our full model.**

|          | CoNLL          | MUC            | $B^3$          | $CEAF_e$       |
|----------|----------------|----------------|----------------|----------------|
| Ours     | 66.71          | 41.43          | 80.32          | 78.38          |
| w/o LDA  | 63.59(-3.12)   | 36.92(-4.51)   | 78.48(-1.84)   | 75.39(-2.99)   |
| w/o SAT  | 64.34(-2.37)   | 38.21(-3.22)   | 78.86(-1.46)   | 75.96(-2.42)   |
| w/o SDT  | 65.46(-1.25)   | 39.82(-1.61)   | 79.54(-0.78)   | 77.04(-1.34)   |

improvements over text-based methods. MAF and WS-MCR are two weakly supervised methods, and both of them exhibit significant improvements in MUC scores compared to other single-modal and multimodal baselines. By employing the semi-supervised learning method and carefully tuning the model with a small amount of labeled data and a large amount of pseudo-labeled data, Semi-MCR achieves the current state-of-the-art (SoTA) results. Most notably, our model significantly surpasses the state-of-the-art by a substantial margin, with a CoNLL F1 improvement of 4.92%, an MUC F1 improvement of 9.57%. Moreover, compared to MAF and WS-MCR, our method demonstrates remarkable enhancements, with MUC F1 improvements of 25.78% and 22.24%, respectively.

### 4.4 Narrative Grounding Results

To evaluate the alignment between image regions and phrases in textual data, we set up a narrative grounding task and conduct a comprehensive comparison between our proposed method existing SoTA approaches, including MAF, WS-MCR, and Semi-MCR. The results in Table 2 demonstrate the superiority of our method. Compared to the current state-of-the-art approach (Semi-MCR), our method achieve a 4.26% and 3.69% improvement in noun phrase and pronoun coreference accuracy, respectively. It is worth noting that our proposed method outperforms the MAF, with a 15.24% and 13.83% increase in noun phrase and pronoun coreference accuracy, respectively. These results highlight the advantages of our method and emphasize the necessity of expanding labeled data and effectively leveraging unlabeled data in scenarios where large-scale annotated data is lacking.

### 4.5 Ablation Study

We perform ablation experiments to evaluate the contribution of each component in our model. As depicted in Table 3, no variant matches the full model's performance, highlighting the indispensability of each component. Specifically, the absence of labeled data augmentation (LDA) results in the most significant performance drop. The CoNLL F1 and MUC F1 scores decreased by 3.12% and 4.51%, respectively, indicating the crucial role of expanding high-quality labeled datasets in increasing data diversity and model performance. To validate the necessity of self-adaptive threshold (SAT), we remove it and set the threshold to 0.9. The sharp decrease in performance (2.37% drop in CoNLL F1) demonstrates the non-negligible impact of self-adaptive threshold in the semi-supervised MCR task. Furthermore, removing the smoothing distance term (SDT) lead to a significant performance drop, indicating the importance of it in reducing noise in unlabeled data and helping the model learn more discriminative feature representations, thus improving the utilization of unlabeled data.

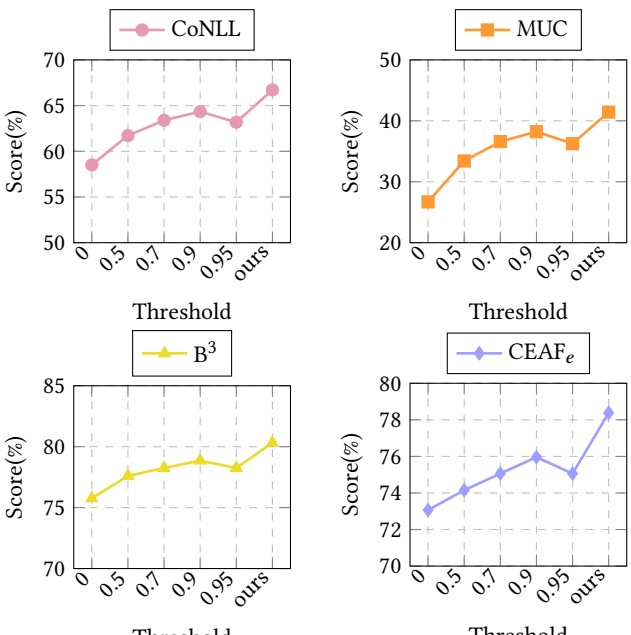

**Figure 4: Comparative results between self-adaptive thresholds and different fixed thresholds.**

### 4.6 Deep Analyses on the Effects of Our Proposed Technologies

To further investigate the effectiveness of our method, we conduct in-depth analyses to answer the following questions, with the aim to deeply mine the intuition and analyze implicit phenomena.

**Q1: What are the advantages of the self-adaptive threshold?** We validate the effectiveness of our self-adaptive threshold by comparing it with different fixed thresholds in Figure 4. The results indicate that the self-adaptive threshold consistently outperforms the fixed thresholds across all four evaluation metrics. This suggests that the self-adaptive threshold strikes a balance between data quality and quantity, enabling a more flexible selection of unlabeled data. Additionally, we observe that when no filtering of pseudo predictions is applied (i.e., setting the threshold to 0.0), the model performs poorly. This highlights the necessity of a threshold-based training strategy. When consistently using lower fixed threshold, such as 0.5, the model shows some improvement compared to including all pseudo predictions, emphasizing the importance of removing noisy samples to a certain extent. While setting higher fixed threshold, like 0.95, the model's performance does not improve compared to a threshold of 0.9, in fact, it even declines. This intriguing phenomenon suggests that excessively filtering out unlabeled data result in the loss of high-quality samples, thereby limiting the model's learning capacity.

**Q2: How to evaluate the quality of generated images?** We conduct a quality evaluation of the images generated by the diffusion model to quantify and assess the quality of the generated images. To find images that better match the caption and evaluate without relying on real reference images and text, we choose two metrics, MUSIQ and CLIPScore, to evaluate the quality of our generated images. MUSIQ is used to assess the quality of the images themselves,

**Table 4: Results of the comparison of the self-adaptive selection strategy with different fixed proportion selection strategies.**

| Training Set | MUC | | | B³ | | | CEAF$_e$ | | | CoNLL |
|---|---|---|---|---|---|---|---|---|---|---|
| | R | P | F1 | R | P | F1 | R | P | F1 | F1 |
| CIN | 37.78 | 38.65 | 36.92 | 72.11 | 86.62 | 78.48 | 63.19 | 94.92 | 75.39 | 63.59 |
| CIN + 25% new | 38.56 | 40.11 | 38.10 | 72.21 | 87.29 | 78.85 | 63.78 | 95.29 | 75.94 | 64.29 |
| CIN + 50% new | 39.01 | 40.79 | 38.72 | 72.29 | 87.74 | 79.09 | 64.20 | 95.61 | 76.36 | 64.72 |
| CIN + 75% new | 39.79 | 43.75 | 40.86 | 72.42 | 89.46 | 79.96 | 65.76 | 96.35 | 77.74 | 66.19 |
| CIN + 100% new | 36.42 | 41.22 | 37.83 | 72.71 | 89.86 | 80.25 | 66.60 | 95.44 | 77.99 | 65.36 |
| CIN + self-adaptive | **39.83** | **44.70** | **41.43** | **72.42** | **90.27** | **80.32** | **66.51** | **96.63** | **78.38** | **66.71** |

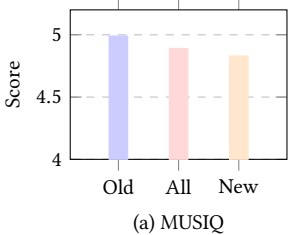
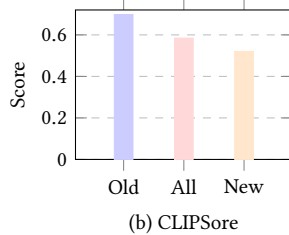

(a) MUSIQ                                (b) CLIPSore

**Figure 5: Generation image quality evaluation. "Old" means the original data, "All" means the original data and generated data, "New" means the generated data.**

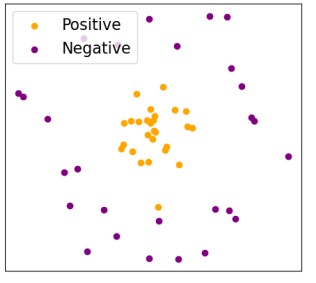
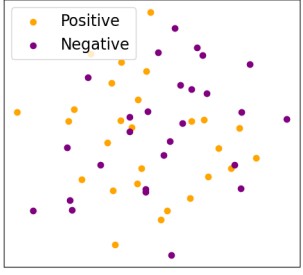

(a) with smooth distance                                (b) without smooth distance

**Figure 6: Visualization of the sample features.**

while CLIPScore is used to evaluate whether the generated images can match and correlate with the given textual descriptions. The evaluation results, as shown in Figure 5, reveal that the generated images (New) achieved high scores on both the MUSIQ and CLIP-Score metrics, approaching those of original images (Old). These results indicate that the generated images are of high quality and align well with the given textual descriptions.

**Q3: What are the advantages of the self-adaptive selection strategy?** To validate the effectiveness of the self-adaptive selection strategy, we compare the results of it with different fixed ratio selection strategies, as shown in Table 4. From the evaluation of image quality, we observe that the overall image quality is high, but there are still a few low-quality images. Thus, we sort the generated images based on their CLIPScore scores and select a portion of them to merge into our training set according to a certain ratio. We find that our self-adaptive selection strategy outperforms all the fixed ratio data selection strategies. We also observe that using all the generated image-caption pairs for the training set does not yield the best results. This might due to the presence of lower-quality images among the generated samples, necessitating the need for filtering based on certain criteria. Additionally, when using lower ratios of generated data, such as 25% and 50%, we notice a slight decrease in overall performance. This suggests that, apart from high-quality samples, the quantity of samples in the training set is also crucial. In conclusion, the above analysis highlight the importance of quality filtering and emphasizes the need to find the optimal balance between high-quality samples and dataset size.

**Q4: What is the role of smooth distance term in performance improvement?** We visualize the sample features using t-SNE to reveal the effectiveness of smooth distance term. In Figure 6 (a), when employing smooth distance term, the feature differences between positive and negative samples are highly evident. This indicates that

the model is encouraged to learn distinct feature representations and achieves better discrimination between positive and negative samples. Conversely, in Figure 6 (b), when smooth distance term is removed, the feature distinctions between positive and negative samples are not significant. This limitation hampers the model's ability to learn discriminative feature representations. This emphasizes the importance of smooth distance term in addressing noise issues in unlabeled data and enhancing model performance.

## 5 CONCLUSION

In this paper, we explore two major challenges in the semi-supervised MCR task: scarcity of labeled data and under-exploitation of unlabeled data, and propose a solution called SLUDA. SLUDA expands high-quality MCR training data through a self-adaptive expansion selection strategy, and makes full use of the unlabeled data through a self-adaptive thresholding strategy. Through experiment evaluation, we have found that all of our claimed innovative approaches and hypotheses have been demonstrated to be effective. One of the most interesting findings is that the performance of the model in the semi-supervised MCR task can be significantly improved by adaptively adjusting the threshold value or ratio according to the learning status of the model, both for unlabeled and labeled data. This finding provides insight into our understanding of the role of data quality and quantity in MCR, and offers great promise for advancing further development of this task.

## ACKNOWLEDGMENTS

This work is supported by the National Key Research and Development Program of China (No. 2022YFB3103602), the National Natural Science Foundation of China (No. 62176187). This research is also supported by the Ministry of Education, Singapore, under its AcRF Tier 2 Funding (Proposal ID: T2EP20123-0052).

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
