# OpenReview forum: "Self-Adaptive Fine-grained Multi-modal Data Augmentation for Semi-supervised Muti-modal Coreference Resolution"
_acmmm.org/ACMMM/2024/Conference — MM2024 Poster_

### Official Review · Reviewer_qX97 · 2024-05-19

**Rating:** 3
**Confidence:** 3

**Summary:**

This paper proposes a self-adaptive fine-grained multi-modal data augmentation framework called SLUDA for semi-supervised multi-modal coreference resolution (MCR). To address the scarcity of labeled data, the authors leverage text coreference resolution datasets and diffusion models to generate fine-grained text-to-image data with aligned text entities and bounding boxes. They introduce a self-adaptive selection strategy to curate the augmented data. To better utilize unlabeled data, they design a self-adaptive threshold strategy that dynamically adjusts the confidence threshold based on the model's learning status. Additionally, a distance smoothing term is incorporated to enhance the discriminative power of the model's feature representations. Experiments on the CIN dataset show that their framework significantly outperforms state-of-the-art baselines.

**Strengths:**

1. The paper addresses important challenges in MCR, namely the scarcity of labeled data and under-exploitation of unlabeled data.

2. The proposed self-adaptive fine-grained data augmentation approach leveraging text datasets and diffusion models is novel and effective for expanding labeled MCR data.

3. The self-adaptive threshold strategy for utilizing unlabeled data based on the model's learning status is a thoughtful approach to balance data quality and quantity.

**Limitations:**

1. The paper lacks a discussion and comparison with recent multi-modal large pretrained models (e.g., CLIP, ALIGN) which have shown strong zero-shot and few-shot capabilities on various multi-modal tasks. It would be valuable to see how the proposed approach compares to or could be combined with such models.

2. While the data augmentation approach is innovative, more details and analyses could be provided. For example, how is the quality of the generated data evaluated quantitatively? How much diversity does it add to the training set? Some visualizations of the generated samples would help readers better understand the approach.

3. The self-adaptive threshold strategy is interesting but lacks ablation studies to validate its effectiveness. It would be good to see how it compares to the fixed threshold baseline and analyze its behavior during training.

4. The distance smoothing term is not clearly formulated in the methodology. More mathematical details are needed to explain how it is incorporated into the loss function and how it affects the feature representations.

**Suitability:**

3

---

### Official Review · Reviewer_vXc1 · 2024-05-21

**Rating:** 4
**Confidence:** 3

**Summary:**

1. The authors propose SLUDA to address the Multimodal Coreference Resolution (MCR) task by enhancing training data from both labeled and unlabeled perspectives. The framework generates fine-grained image-caption pairs and selects high-quality data for labeled data augmentation, while a self-adaptive threshold strategy and distance smoothing term are used to utilize unlabeled data effectively.

2. The SLUDA framework employs two modality-specific Variational Autoencoders (VAEs) to learn latent representations for different modalities. Experimental results on the CIN dataset show that SLUDA achieves state-of-the-art performance, significantly outperforming existing baselines. It improves the MUC F1 score by at least 9.57% and the CoNLL F1 score by 4.92%, with a remarkable 22.24% enhancement against weakly-supervised baselines.

**Strengths:**

1. The idea of using self-adaptive methods for both labeled and unlabeled data augmentation is really impressive. It’s like the framework is smart enough to know what data to use and how to use it, ensuring the quality and diversity of the training data (which is crucial for effective learning).

2. The performance improvements are quite remarkable. Achieving over a 22% boost in the MUC F1 score compared to weakly-supervised baselines is no small feat. This shows that the framework doesn't just look good on paper but actually delivers significant real-world benefits (which is what we all hope for in new methodologies).

**Limitations:**

1. In section2.2, "labeled data may not always be sufficient due to the 224 cost of annotation", there may lack of related work to present in Section 2.2 , especially considering the task of " tapping into the unlabeled data" in the first stage of this paper:

[1] Towards Language Models That Can See: Computer Vision Through the LENS of Natural Language

[2] S-clip: Semi-supervised vision-language learning using few specialist captions

[3] Gpt self-supervision for a better data annotator

[4] OpenAnnotate2: Multi-Modal Auto-Annotating for Autonomous Driving

2. The framework relies heavily on synthetic data generation for labeled data augmentation. While this approach helps mitigate the issue of scarce labeled data, it might not fully capture the complexity and variability of real-world data. This could limit the model’s generalizability when applied to entirely new or diverse datasets.

3. The use of self-adaptive techniques and multiple VAEs adds a layer of complexity to the implementation. This could pose challenges for practitioners who might not have the resources or expertise to replicate the results.

**Suitability:**

2

---

### Official Review · Reviewer_xjND · 2024-05-27

**Rating:** 5
**Confidence:** 3

**Summary:**

This paper focuses on multi-modal corresponding resolution (MCR) tasks. Due to the data scarcity issues, the authors propose a self-adaptive fine-grained multi-modal data augmentation method by  utilizing diffusion model to generate synthetic labeled data based on a few labeled data and exploiting MCR model trained on both original and synthetic labeled data to generate pseudo labels on unlabeled data. They evaluated their proposed method on CIN datasets and verified the effectiveness of this method.

**Strengths:**

1. Although the basic idea is not very novel in data augmentation methods, its application on multi-modal data still needs more efforts.
2. The paper is clearly written and well organized.

**Limitations:**

1. There are only one dataset used in the experimental evaluation, which can not fully verify the effective of this method on other datasets. More benchmarks would be better.
2. Since the MCR task is relatively complex, a direct analysis (e.g., case study with both image and text, error analysis) is necessary to provide more insights for future researches.

**Suitability:**

3

---

### Meta-Review · Area_Chair_y2uq · 2024-07-01

**Recommendation:** Accept (Poster)
**Confidence:** 4

**Metareview:**

This paper introduces a self-adaptive fine-grained multi-modal data augmentation method (SLUDA) to improve semi-supervised multi-modal coreference resolution (MCR) by generating synthetic labeled data and leveraging both labeled and unlabeled data. The approach effectively tackles data scarcity and enhances model performance on the CIN dataset, showing significant improvements over existing baselines. The strengths of the paper include its innovative application of data augmentation techniques to multi-modal data, clear organization, and impressive performance gains. However, the evaluation is limited to a single dataset, and the paper would benefit from additional benchmarks to generalize its effectiveness. Furthermore, while the data augmentation approach is creative, it lacks detailed analysis and visualizations of the generated data. The self-adaptive threshold strategy and distance smoothing term are promising but require more rigorous validation and clearer formulation. Addressing these concerns in the camera-ready version by including more datasets, comprehensive analysis, ablation studies, and comparisons with recent large pre-trained multi-modal models would strengthen the paper's contributions.